# OpenReview forum: "Distributive Fairness in Large Language Models: Evaluating Alignment with Human Values"
_NeurIPS.cc/2025/Conference — NeurIPS 2025 poster_

### Official Review · Reviewer_AP6N · 2025-06-29

**Clarity:** 3
**Significance:** 2
**Originality:** 2
**Rating:** 4
**Confidence:** 4

**Summary:**

The paper investigates whether recent LLMs (GPT-4o, Claude-3.5, Llama-3-70B, Gemini-1.5) produce resource-allocation decisions that align with human notions of distributive fairness. Using a classic benchmark from experimental-economics literature (Herreiner and Puppe, 2010), the authors compare human answers against 100 answers/generations per LLM. The authors analyze which fairness notions (Equity, Envy-Freeness, Rawls-Maximin, USW, Pareto) each answer satisfies and explore the use of money as a redistributive tool. Main findings: (i) models favor efficiency (PO) and EF, largely ignoring perfect equity; (ii) only GPT-4o somewhat uses money to reduce inequality; (iii) when forced to choose from a menu, GPT-4o & Claude shift more towards equitable allocations.

**Questions:**

- Each model was queried 100 times per question/instance; Were all generations perfectly valid for all models? How was valid output ensured? What was the percentage of invalid generations?
(Other questions in the "Strengths & Weaknesses" section)

**Ethical Concerns:**

["NO or VERY MINOR ethics concerns only"]

**Final Justification:**

Authors have partially addressed my concerns. The paper would definitely be stronger with the promised additions.

**Limitations:**

Limitations are discussed in relation to the human study by Herreiner and Puppe, but somewhat ignored in relation to the current paper's  claims could be dependent on the specific dataset that was used.

**Paper Formatting Concerns:**

- Figure 2 and similar figures are a bit confusing; it might be better to use colors corresponding to different fairness notions, instead of having the (color, fairness-notion) map change for each plot;

**Quality:**

2

**Strengths And Weaknesses:**

**Strengths**
- The specific application to _distributive_ fairness is the main novelty of the paper. It is an interesting contribution, as LLMs are bound to be used to allocate resources (either by experts or laymen).
- The paper has extensive robustness checks (e.g., varying order of prompts).

**Weaknesses**
- There is a variety of other “LLM-alignment” papers already out there:
  - Related work should definitely be extended beyond the current 1/3rd of a page. The current related work addresses the use of LLMs as social/economic agents, but it does not address related work on LLM-alignment or LLM fairness.
  - Afterwards, the paper should clearly state its contributions and exactly how it differs from similar works in the literature;
- I suspect any LLM-generated resource allocation will be greatly dependent on the specific prompt and context instructions provided to the model. Given that the main contribution of the paper is the “distributive fairness” setting, further results on how different prompts or workflows affect allocations would improve the paper. For example, providing explicit definitions of fairness (words like “minimizing inequality” are subjective to LLMs and to humans). Does LLM resource allocation depend on the _type_ of resources being allocated? (e.g., models could behave differently for medical resources compared to less critical resources).
- Seeing results on one or more datasets would be more convincing.
- Some form of confidence interval should be added to all numeric results, given that these are sampled with n=100 for each LLM-question pair;

---

> ### Author Rebuttal · Authors · 2025-07-30
>
> We thank the reviewer for the constructive comments in improving the paper. We will incorporate your suggestion about the related work and discuss how the current paper differentiates itself. Below, we address the comments and questions raised by the reviewer.
>
> > I suspect any LLM-generated resource allocation will be greatly dependent on the specific prompt and context instructions provided to the model. Given that the main contribution of the paper is the “distributive fairness” setting, further results on how different prompts or workflows affect allocations would improve the paper. For example, providing explicit definitions of fairness (words like “minimizing inequality” are subjective to LLMs and to humans). Does LLM resource allocation depend on the type of resources being allocated? (e.g., models could behave differently for medical resources compared to less critical resources).
>
> **Design choices for evaluating alignment:** We intentionally argue about misalignment between humans and LLMs based on LLMs’ responses to a prompt adapted as closely as possible to the instructions given to humans. These instructions, which requires the decision-maker to “determine the allocation you consider to be the fairest”, involves judgements about the definition of the word “fairest”. The goal is to identify whether LLMs represent the (often diverse) distributive preferences of humans—something that requires the interpretation of what fairness means.
>
>
>
> **Ability to ensure specific fairness properties:** We also report LLMs’ behavior when given prompts that include specific definitions of fairness (section 5 and Appendix F.2). We find that LLMs struggle with computing allocations satisfying specific notions of fairness, even with unambiguous instructions such as “Your task is to determine the allocation in which all individuals have exactly the same value for their respective bundles”, or “Your task is to determine the allocation where each individual prefers their own bundle the most”. These observations raise questions about the ability of commonly used chatbots to provide allocations that serve the purpose for users who explicitly provide their distributive preferences.
>
>
>
> **Sensitivity to context and prompting:** Furthermore, we demonstrate how the context provided and the framing of the prompt can heavily impact LLMs’ responses, as the reviewer correctly identifies. We show how LLMs’ preferences change significantly when they are assigned the role of a recipient (section 5). We also show how LLMs’ become biased to welfare maximization when the intention is modified, i.e. where the goal is to provide an allocation that is the “most desirable” or “acceptable by all” instead of the “fairest” (section 5 and Appendix F.1). Similarly, providing LLMs with options to choose from, and context about human preferences (section 4.1), leads to a change in preferences as compared to the case where they are required to generate allocations from scratch without any context about allocations humans find desirable.
>
>
>
> **Extensions to other settings:** Given the degree of variation in LLMs’ distributive preferences we observe due to modifications to the instructions and context provided, we suspect there to be substantial variations in different resource allocation settings based on factors such as the (i) type of resource being allocated (e.g. medical resources vs. luxury items), or (ii) the relationship between the decision-maker and recipients (e.g. friends vs. competitors), etc. While these are interesting directions for future work, our paper aims to identify the challenges with using LLMs for resource allocation, due to their inability to fully represent human preferences and sensitivity to exact instructions and context provided.
>
>
> > Each model was queried 100 times per question/instance; Were all generations perfectly valid for all models? How was valid output ensured? What was the percentage of invalid generations?
>
> We used a two-staged prompting strategy (described in detail in Appendix J) to ensure the validity of each sample collected. The first stage involved generating a free-text response from the LLM, without enforcing any formatting constraints. In the second stage, the model was asked to arrange the allocation it provided in the free-text response as a JSON object. We manually verified the validity of the second step, for a subset of samples, in correctly identifying the allocation returned by the LLM. The allocation indicated by LLMs in the first step were misidentified in the second step in less than 1% samples.
>
> In terms of invalid generations, we did not find any responses that deviated from the instructions (to indicate an allocation) in instances involving only goods. The only type of invalidity we observed was in instances with money, where LLMs tended to allocate more money than was available, although this behaviour was limited to ~2% responses.
>
> > Some form of confidence interval should be added to all numeric results, given that these are sampled with n=100 for each LLM-question pair;
>
> Thank you for the valuable suggestion. We will include confidence intervals (or a similar measure of deviation) wherever we report aggregate metrics for our experiments. For each numeric result, we use Fisher’s Exact test, and consider any difference as significant if the p-value is less than 0.05. This is true for comparisons between LLMs and humans, or any two LLMs, both at the level of individual instances and at the aggregate level. For example, the statement that Claude-3.5-S returns envy-free (EF) allocations significantly more frequently than humans do, is based on a statistical comparison (using Fisher’s Exact test) of the fraction of responses from humans and Claude where EF is satisfied.

---

> > ### Comment · Reviewer_AP6N · 2025-08-05
> >
> > Thank you for the clarifications; most factual questions are now cleared up. After reading the rebuttal as well as others' reviews, a few important points still need attention:
> >
> > - Related work should definitely be extended beyond the current 1/3 of a page, as pointed out by other reviewers as well. Cover recent LLM-alignment/prosocial-behaviour studies and the wider behavioural-economics literature on fairness, not just Herreiner&Puppe. The paper results are interesting but are not properly contextualized within the existing literature, and this would be necessary to properly judge the results.
> >
> > - Precisely define what "alignment" means when human responses are heterogeneous, and use this benchmark consistently.
> >
> > - Summarise prompt sensitivity with a small quantitative table so readers can see how model choices shift across variants and note that the equitability gap remains.
> >
> > - Add basic statistical reporting: validity-rate numbers (~1%), confidence intervals or standard errors.
> >
> > - Explicitly label domain coverage as a limitation: Human fairness norms differ by resource type, so deferring medical or public-goods scenarios to future work should be acknowledged.
> > 	- In fact, the paper claims to answer "Do LLMs act in alignment with human and societal values in resource allocation tasks?", but humans will definitely distribute different resources differently (this fact in itself is also missing  discussion on related work).
> >
> > - It would be interesting to bring one clear failure case (where models miss an explicit fairness instruction) into the main text for transparency.

---

> > > ### Author Response · Authors · 2025-08-05
> > >
> > > We sincerely thank the reviewer for their suggestions and will certainly incorporate these into the final version of the paper, by making use of the extra space provided for the camera-ready version. This shall include:
> > > - An **expanded related-work** section, which addresses how our findings related to the literature on social preferences and behavior, and perceived fairness in both humans and LLMs. Here, we shall also incorporate a discussion on how perceived fairness changes across **different resource allocation settings** (based on the type of resource, relationship between stakeholders, etc.).
> > > - A clearer discussion on the definition and usage of the word **"alignment"**.
> > > - A brief tabular summary of the results pertaining to **non-semantic prompting changes** (prompt sensitivity).
> > > - A clearer description of **statistical comparisons** (with confidence intervals) and **validity rates**.
> > > - An illustration of the **errors** made by LLMs while trying to **optimize** for specific fairness notions (e.g. equitability), similar to the example provided in Appendix K).

---

> > > > ### Comment · Reviewer_AP6N · 2025-08-07
> > > >
> > > > Thank you for your efforts. The promised additions would definitely improve the paper.
> > > >
> > > > I still think that such large modifications would need a review beyond the comment section of a rebuttal; but will raise my suggestion to a borderline accept.

---

> > > > > ### Author Response · Authors · 2025-08-08
> > > > >
> > > > > We are glad that our response helps resolve any issues and we appreciate your decision to raise the score.

---

### Official Review · Reviewer_hTpc · 2025-06-30

**Clarity:** 2
**Significance:** 3
**Originality:** 3
**Rating:** 4
**Confidence:** 3

**Summary:**

The paper addresses whether large language models (LLMs) align with human notions of distributive fairness. Using benchmark examples from resource-allocation problems and variants, the authors compare how LLMs allocate resources to humans across fairness dimensions: equitability, envy-freeness, Rawlsian ideas, Pareto optimality, and utilitarian social welfare. They find that, unlike humans who  minimize inequality, LLMs mostly favor efficiency or envy-freeness. The allocations they produce are usually not equitable, and GPT-4o is the only model that, when given the option to use money to more finely divide allocations, actually reduces disparities. Overall, the work presents a benchmark and highlights differences between current LLM behavior and human distributional preferences.

**Questions:**

- What do the results look like for more models, especially reasoning models?
- How do you distinguish between genuine differences in the models' distributive principles from artifacts of prompt framing?
- When a model's allocation deviates from an equity-focused rule, how do you differentiate its goals from its results?

**Ethical Concerns:**

["NO or VERY MINOR ethics concerns only"]

**Final Justification:**

Thank you for the detailed rebuttal and the additional results. I will maintain my original (positive) score

**Limitations:**

Yes

**Quality:**

3

**Strengths And Weaknesses:**

Strengths:
- Research question: The paper addresses an interesting (and impactful) question: whether LLMs output results that align with human notions of distributive fairness. The consequences of the findings are clear.
- Methodological design: The way that the problem is formulated (using examples from the experimental economics literature) is interesting and the testbed allows for comparing many kinds of fairness. It's a nice way to do it, especially because we have comparisons for people.
- Analysis: The analysis is interesting and robust. For example, the work on investigating model robustness is interesting and this kind of stress testing is quite important.

Weaknesses:
- Number of models: The experiments only consider four LLMs: GPT-4o, Claude-3.5 Sonnet, Llama-3 70B, and Gemini-1.5 Pro. To reach general conclusions about LLMs it would be more compelling to see results for more models and especially mode advanced models (e.g. reasoning models).
- Prompt sensitivity: The findings about prompt sensitivity are interesting (e.g. minor changes such as requiring a JSON answer flip the modal output in many instances). If preferences swing with formatting, doesn't it invalidate the idea that LLMs systematically allocate resources differently from people? Instead it seems that  models lack coherent distributive principles.
- Related to the above, it's possible that a model has a "goal" but just isn't able to compute the allocation that results in that goal. E.g. if an allocation problem requires doing long division, models (especially the non-reasoning models) may not be able to do it. It doesn't seem we should infer that models aren't optimizing for equity here if this is the case. Broadly, I think it makes sense to separate a model's "goals" from what it actually produces.

---

> ### Author Rebuttal · Authors · 2025-07-30
>
> Thank you for the constructive comments and interesting questions. We shall answer them below:
>
> > What do the results look like for more models, especially reasoning models?
>
> We conducted our experiments with Gemini-2.5-Pro (a SOTA reasoning model), with which our primary results still hold. Here is a summary of our findings:
>
> 1.  The model continues to ignore equitable (EQ) allocations. Its responses are much more deterministic (in terms of the allocation returned for a given instance), and prioritizes notions such as envy-freeness (EF) and Pareto-optimality (PO) instead.
>
> 2.  In instances with money, only when EQ allocations are also Pareto-optimal, Gemini-2.5-Pro returns such allocations. However, this seems to be primarily motivated by PO, since the model never returns EQ allocations that are not PO. This behaviour is different from the non-reasoning models, potentially due to the reasoning model’s enhanced ability to satisfy additional desirable properties along with the properties it considers most important.
>
> 3.  Gemini-2.5-Pro does not choose EQ allocations among a set of options (unlike GPT-4o and Claude-3.5-Sonnet). Instead, it selects allocations satisfying EF or PO (or both). This is true even in instances with money, where it returns EQ+PO allocations when required to generate allocations from scratch (as described above). This adds to the evidence that the reasoning model does not prioritize equitability.
>
> 4.  When explicitly instructed to compute EQ allocations, Gemini-2.5-Pro can do so with over 95% accuracy, where non-reasoning LLMs do not cross 18%. Given the former’s enhanced reasoning capabilities, this is expected. However, this further strengthens the observation that Gemini-2.5-Pro ignores EQ allocations not because it struggles to compute them (unlike for the non-reasoning LLMs), but because it may not equate “fairness” with “equitability”.
>
> Hence, the difference between the perceived fairness of humans and the reasoning model is even clearer (given that it does not select EQ allocations among a set of options). We will be updating the paper with these results.
>
> > How do you distinguish between genuine differences in the models' distributive principles from artifacts of prompt framing?
>
> The ground truth of LLMs’ distributive preferences that we use for comparing with humans is based on free-flow responses (without any enforced formats) to instructions adapted (as closely as possible) from those given to human respondents. This comparison reveals certain clear differences between the preferences of humans and those of LLMs, such as the neglect of equitability and enhanced preference for economic efficiency from the latter.
>
> As the reviewer correctly points out, our experiments show that non-semantic changes to the prompt (i.e. those where the underlying task is the same) lead to significant changes in the distribution of responses (Appendix D). This reveals the lack of consistent distributive preferences in LLMs. However, across all such changes, the misalignment between humans and LLMs only increases, with an increased share of responses for efficiency and no increase in the share of responses ensuring equitability, from LLMs. While mapping the exact distributive preferences of LLMs’ across diverse settings is a challenging task that requires extensive future work, our work identifies key areas of misalignment between humans and LLMs that persist across different prompt framings.
>
> > When a model's allocation deviates from an equity-focused rule, how do you differentiate its goals from its results?
>
> This is an excellent and profoundly important question: How can we disentangle and differentiate between an LLM’s intentions, abilities, and underlying values?
>
> Precisely to understand this, we experiment with a variation of the task, where the LLMs need to select the allocation they consider fairest among a set of options satisfying different notions (section 4.1). Our results show that two out of the four LLMs (GPT-4o and Claude-3.5-Sonnet) that we test select EQ allocations among a set of options although they fail to return EQ allocations when required to generate their preferred allocation from scratch. As correctly pointed out by the reviewer, a potential reason for this is the inability to compute EQ allocations even when specifically asked to do so—something we shown in Section 5.
>
> However, as we show in Section 4.2, using reasoning-enhancement prompting techniques such as Chain-of-Thought does not uniformly improve alignment. Similarly, using an LLM with advanced reasoning capabilities (such as Gemini-2.5-Pro) does not ensure alignment either, since such models may have different goals altogether (as discussed above).

---

> > ### Comment · Reviewer_hTpc · 2025-08-04
> >
> > Thank you for the detailed rebuttal and the additional results. I will maintain my original (positive) score

---

### Official Review · Reviewer_e8mE · 2025-06-30

**Clarity:** 3
**Significance:** 3
**Originality:** 3
**Rating:** 4
**Confidence:** 3

**Summary:**

This paper studies the default fairness interpretation of LLMs and how this interpretation aligns or differs from that of humans (as understood through the lens of a prior work). In particular, the authors use the settings from Herreiner and Puppe for different possible ways of allocating indivisible goods. Different allocations correspond to different notions of fairness and/or economic efficiency (utilitarian social welfare, Pareto optimality, equitability, envy-freeness, minimax etc). When LLMs are prompted to generate a "fair" allocation, the authors observe that LLMs prioritize economic efficiency, compared to humans (in Herreiner and Puppe study) who give more consideration to equitability. This is the main result of the paper. Then there are some other results some of which are clear and some aren't that clear. For example, when asked to select from a list of option (instead of generating answer), LLM behavior shows a significant difference (not they do prioritize equitability). The effect of prompting, assigning personas etc do not show clear trend, but these are interesting experiments nevertheless.

**Questions:**

Please see the questions asked in the weakness section above.

**Ethical Concerns:**

["NO or VERY MINOR ethics concerns only"]

**Final Justification:**

After reading all the other reviews and authors' rebuttal, I keep the original overall score (i.e. borderline accept).

**Limitations:**

Yes,

**Quality:**

3

**Strengths And Weaknesses:**

Strengths:

1. Understanding LLM behavior and their default preferences (including their steerability) is an important subject. While there are a lot of papers study gender/racial bias in LLMs, this paper studies fairness in resources allocation scenarios and fairness from social choice perspective.

2. There are quite a few different experiments performed (generation, selection from different menus, effect of prompting, effect of assigning personas) etc, that make the paper overall sufficiently complete.

I am positive about this paper. I am inclined to further increase my overall score after reading authors' response and other reviews.

Weaknesses:

1. Many of the claims in this paper about human preferences are based on a single citation (i.e. Herreiner and Puppe). I wonder if the authors can cite additional papers to support this. I am not fully convinced about the claims regarding human preferences.

2. The second concern is that there may be potential confounders (e.g. the reasoning of LLMs or any errors in their calculations while deciding the "fairness score" of different allocations). This concern doesn't make the results invalid but it would be interesting to decode what exactly is going on and whether we are reading too much into these results. Have the authors tried prompting the LLMs to explain their reasoning while generating or selecting answers? The generated reasoning will be more difficult to evaluate automatically or manually but some random samples can be evaluated and included in the paper for a more qualitative insight.

3. Related to the above, it would be useful to use some reasoning models (e.g. R1 or o1) also in these experiments and see whether the results change.

---

> ### Author Rebuttal · Authors · 2025-07-30
>
> We thank the reviewer for the constructive comments in improving the paper. Below we address the questions raised by the reviewer.
>
> > Many of the claims in this paper about human preferences are based on a single citation (i.e. Herreiner and Puppe). I wonder if the authors can cite additional papers to support this. I am not fully convinced about the claims regarding human preferences.
>
> To support our claim that equitability, often referred to as “inequality-aversion”, is prioritized by humans in distributive settings, we cite a variety of seminal works in our paper (line 184). In addition to these, a large body of work shows how inequality aversion is a key factor in the distribution [1] and re-distribution of income [2], dispute resolution [3], and healthcare policy [4]. In fact, even experiments using functional MRIs of human subjects involved in experiments involving monetary exchanges provide evidence for human sensitivity to inequality [5].
>
> [1] Choshen-Hillel, S., & Yaniv, I. (2011). Agency and the construction of social preference: Between inequality aversion and prosocial behavior. Journal of Personality and Social Psychology, 101(6), 1253–1261
>
> [2] Dawes, C., Fowler, J., Johnson, T. et al. Egalitarian motives in humans. Nature 446, 794–796 (2007).
>
> [3] Loewenstein, G. F., Thompson, L., & Bazerman, M. H. (1989). Social utility and decision making in interpersonal contexts. Journal of Personality and Social Psychology, 57(3), 426–441.
>
> [4] Costa-Font, J., Cowell, F. Incorporating Inequality Aversion in Health-Care Priority Setting. Soc Just Res 32, 172–185 (2019).
>
> [5] Tricomi, E., Rangel, A., Camerer, C. et al. Neural evidence for inequality-averse social preferences. Nature 463, 1089–1091 (2010).
>
> > The second concern is that there may be potential confounders (e.g. the reasoning of LLMs or any errors in their calculations while deciding the "fairness score" of different allocations). This concern doesn't make the results invalid but it would be interesting to decode what exactly is going on and whether we are reading too much into these results. Have the authors tried prompting the LLMs to explain their reasoning while generating or selecting answers? The generated reasoning will be more difficult to evaluate automatically or manually but some random samples can be evaluated and included in the paper for a more qualitative insight.
>
> We indeed analyzed the explanations from the LLMs for various questions and found some interesting insights. The primary observation is that LLMs use a “greedy approach” to allocate resources. This involves either (i) a round-robin-like mechanism where agents are picked one-by-one and allocated their favorite good among the remaining ones, or (ii) assigning goods one-by-one to individuals who value them the most. The first procedure often leads to allocations satisfying envy-freeness and the second ensures maximum social welfare, which concurs with observations from our quantitative analysis that LLMs return such allocations frequently. On the other hand, such procedures would rarely result in equitable allocations (especially in the instances we consider). Instead, computing an equitable allocation would require assigning goods to individuals who do not value them the most—something LLMs seem reluctant to do.
>
> Furthermore, in instances with money, most LLMs fail to identify that allocating the money to a single agent can make the allocation envy-free or equal overall, and instead attempt to allocate even the money equally, which results in no fairness property being satisfied.
>
> We appreciate the feedback, and will definitely include such insights from qualitative analysis of the explanations in our paper.
>
> > Related to the above, it would be useful to use some reasoning models (e.g. R1 or o1) also in these experiments and see whether the results change.
>
> We are happy to share results from recent experiments we conducted with Gemini-2.5-Pro, a SOTA reasoning model.
>
> 1.  Equitable (EQ) allocations are still rarely returned. While the responses from Gemini-2.5-Pro are much more deterministic (in terms of the allocation returned for a given instance), the model ignores EQ allocations, and prioritizes notions such as envy-freeness (EF) and Pareto-optimality (PO) instead.
>
> 2.  In instances with money, only when EQ allocations are also Pareto-optimal, Gemini-2.5-Pro returns such allocations. This seems to be primarily motivated by PO, since in cases where EQ allocations are not PO, the model never returns them. This difference between the reasoning model and the non-reasoning models is potentially due to the reasoning model’s enhanced ability to satisfy additional desirable properties along with the properties it considers most important.
>
> 3.  Gemini-2.5-Pro does not select EQ allocations from a menu of options (unlike GPT-4o and Claude-3.5-Sonnet). Instead, it chooses allocations satisfying EF or PO (or both). Surprisingly, this is true even in instances with money, where it returns EQ+PO allocations when required to generate allocations from scratch. This strengthens the observation that the reasoning model does not prioritize equitability.
>
> 4.  Unlike the non-reasoning LLMs, Gemini-2.5-Pro can compute EQ allocations with more than 95% accuracy, when instructed explicitly to compute one. Given the latter’s enhanced reasoning capabilities, this is expected. However, this further strengthens the observation that Gemini-2.5-Pro ignores EQ allocations not because it struggles to compute them (which is the case with the other models), but because it may not equate “fairness” with “equitability”.
>
> Hence, the main results, regarding the misalignment between humans and LLMs regarding equitability, hold even with an advanced reasoning model like Gemini-2.5-Pro. If at all, the difference between the perceived fairness of humans and the reasoning model is even clearer (given that it does not select EQ allocations among a set of options). We will be adding these results to the paper.

---

### Official Review · Reviewer_vXHW · 2025-07-02

**Clarity:** 3
**Significance:** 3
**Originality:** 2
**Rating:** 4
**Confidence:** 4

**Summary:**

The paper studies how various large language models (LLMs) make allocation decisions involving finite resources across agents, using fairness as the criterion. It compares these model-generated decisions to those made by human participants in similar experimental setups. The study finds that LLMs’ decisions often diverge from human judgments and that model responses are sensitive to framing, for example, whether they must choose from a predefined menu or respond while adopting a particular persona.

**Questions:**

1.	Clarify what benchmark is used to determine “alignment” with humans: The paper reports that LLMs deviate from human responses, yet the human data itself shows significant heterogeneity. Meanwhile, models like GPT-4o and Claude select EQ allocations over 60% of the time in the menu setting. Are these models misaligned because they differ from the most common human response, or overly aligned with a narrow fairness principle? Please clarify how alignment is defined and assessed across different settings.
2.	I recommend being more explicit in the text with the prompt phrasing for each experiment, especially in Section 4.1: Was the model asked to “choose the fairest allocation” or simply to “choose one”? This distinction matters a great deal, as it shapes whether the task is interpreted as a normative judgment, a preference prediction, or something else.
3.	Discuss the choice of sampling temperature and implications of repeated querying. At a high temperature, model responses may be highly stochastic, raising questions about whether observed variation reflects noise or stable preferences. Would lower temperature settings produce more consistent or more human-aligned responses? A sensitivity analysis would help.
4.	Clarify the interpretation of the “Cognitive Bias” results.
5.	The paper would benefit from a more in-depth comparison with the growing literature on LLMs and prosocial behavior, economic games, and normative reasoning. While some of this work is mentioned in passing on page 3, the paper misses the opportunity to position its findings in relation to this broader body of research.

**Ethical Concerns:**

["NO or VERY MINOR ethics concerns only"]

**Final Justification:**

I appreciate the engagement during the rebuttal period but my main concern still stands. The same way a paper comparing responses between two treatments in an experiment requires a clear articulation of the test used to reject the null hypothesis, the same is needed when comparing human and AI responses beyond just pointing out that they are not identical. I therefore maintain my score.

**Limitations:**

yes

**Quality:**

3

**Strengths And Weaknesses:**

Strengths-

-	The paper adds to the existing literature that compares human preferences, particularly in experimental settings to LLM responses, by focusing on allocation decisions that may tradeoff different normative criteria.
-	The paper does a nice job at following the work Herreiner & Puppe (2007) study by applying their solicitation method as well as testing other framings that impact LLM choices. The paper goes beyond simple fairness labeling and explores how model behavior changes with prompt variation, personas, and menu constraints
-	The evaluation covers multiple state-of-the-art LLMs (GPT-4o, Claude-3.5S, Llama3-70B, Gemini-1.5P), offering a comparative view of fairness-related behavior across models.
-	I find the results of the significant disagreement across models, even when given a closed set of options, really interesting.


Weaknesses-

-	The paper frequently claims that LLMs are not aligned with human preferences, yet the human data itself shows substantial heterogeneity. For example, only 12.4% of human responses select perfectly equitable EQ allocations. Without a clear normative or empirical benchmark for what counts as “alignment,” it's unclear how to interpret these results. Ironically, LLMs may be even more misaligned in the menu-based tasks, where GPT-4o and Claude select EQ 60–80% of the time, far more than any human subgroup. A more precise definition of alignment is needed to support the paper’s claims.
-	In the original human study, participants selected an allocation from the full feasible space, not from a small curated menu. The LLMs’ strong preference for EQ in the menu setting may partly reflect the salience or availability of that option. Without a directly comparable human menu task, it is difficult to draw conclusions about whether menu-based responses represent improved alignment or distorted heuristics. Overall, it is confusing as to whether the paper is about testing what makes LLMs prefer EQ distributions or about what makes LLMs align more closely with humans.
-	The models are queried 100 times per instance at temperature 1, but the paper does not explain why this sampling procedure is appropriate or what impact it might have on the results. At high temperature, models may behave inconsistently or randomly, which could undermine conclusions about principled misalignment. The paper does not report variance, entropy, or mode frequencies, making it hard to assess whether models are consistent or not in their behavior. I kept on asking myself whether I would expect greater or lower alignment with human experiments with a higher temperature.
-	Especially in the menu-based experiments (Section 4.1), the paper does not include the exact prompt used, leaving it ambiguous whether models were instructed to select the “fairest” allocation or simply “choose one.” This ambiguity makes it difficult to interpret model behavior- are the models expressing fairness judgments, simulating human preferences, or selecting based on other heuristics? Also note that the “Cognitive Bias” section was extremely hard to follow.
-	Beyond noting which fairness notions each model’s responses satisfy, the paper doesn’t measure internal consistency (e.g., are models always returning the same type of solution for a given problem?), or how certain the models are in their outputs. Especially in high-variance tasks like these, it would help to distinguish between principled disagreement and noise.
-	Insufficient engagement with related literature- there is now a large and growing body of work comparing LLM behavior to human experimental results, especially in prosocial and economic game contexts. While some of this literature is briefly cited on page 3, the paper does not meaningfully engage with it. This is surprising given the close methodological parallels. A more detailed comparison of how the present findings align with or diverge from previous studies, and why, was very much needed.

Also, here are a few additional papers on LLM prosocial behavior that the authors may want to look at-

https://www.accessecon.com/Pubs/EB/2024/Volume44/EB-24-V44-I1-P3.pdf

https://www.nature.com/articles/s41598-024-73306-x

https://arxiv.org/abs/2301.02330

---

> ### Author Rebuttal · Authors · 2025-07-30
>
> We thank the reviewer for their fruitful and constructive comments. The answers to the reviewers questions are given below:
>
> > Clarify what benchmark is used to determine “alignment” with humans: The paper reports that LLMs deviate from human responses, yet the human data itself shows significant heterogeneity. Meanwhile, models like GPT-4o and Claude select EQ allocations over 60% of the time in the menu setting. Are these models misaligned because they differ from the most common human response, or overly aligned with a narrow fairness principle? Please clarify how alignment is defined and assessed across different settings.
>
> Alignment is evaluated by comparing the **distribution of LLM-generated allocations** against the **empirical human distribution**, with heterogeneity in human data accounted for by comparing choice frequencies across multiple fairness notions. In menu-based settings, the analysis aims to uncover the source of misalignment: is it due to LLMs prioritizing welfare maximization, computational constraints in exploring many possibilities, error in calculations, or reasoning errors such as losing track in their chain of thought?
>
> The fact that models like GPT‑4o and Claude select equitable (EQ) allocations over 60% of the time indicates that they inherently value fairness notions like EQ when presented with explicit options (and other models do NOT). This observation suggests that, although LLMs remain misaligned with the diverse human distribution, their behavior can potentially be guided and refined to achieve better alignment.
>
> > I recommend being more explicit in the text with the prompt phrasing for each experiment, especially in Section 4.1: Was the model asked to “choose the fairest allocation” or simply to “choose one”? This distinction matters a great deal, as it shapes whether the task is interpreted as a normative judgment, a preference prediction, or something else.
>
> As demonstrated in the sample prompt given in Appendix J.5, the instruction is to “...indicate the allocation you think is fairest…”. This instruction is used to make sure that the only difference between the original prompt (the questions asked to humans) and the prompt used in section 4.1 is the addition of the set of options to choose from. The reviewer correctly points out that the precise question can have a significant impact on the distribution of responses—something we address in Appendix F.1. We appreciate the feedback on being more explicit in the main text about the prompt framing, and shall incorporate the same into the final version.
>
> > Discuss the choice of sampling temperature and implications of repeated querying. At a high temperature, model responses may be highly stochastic, raising questions about whether observed variation reflects noise or stable preferences. Would lower temperature settings produce more consistent or more human-aligned responses? A sensitivity analysis would help.
>
> We select temperature 1 as it is the default setting for most models, and end users typically do not adjust this parameter. Hence, our evaluation reflects the behavior that is most likely to be observed in real-world usage, where the default temperature is effectively the standard operating condition.
>
> Based on preliminary experiments with a temperatures of 0 and 2, we observe that alignment with humans does not improve. This is primarily for two reasons, (i) for temp = 0, the outputs of the LLMs are more deterministic and fail to capture the diversity of human preferences (which is done to a better extent with temperature=1), and (ii) equitable allocations are still not returned at either temperature. Rather, there is no clear difference between the distributions returned at temperatures of 1 and 2. We will add more details on the temperature sensitivity analysis to the paper.
>
> > Clarify the interpretation of the “Cognitive Bias” results.
>
> The “Cognitive Bias” results examine whether LLMs replicate or diverge from common human cognitive biases, such as framing effects or bystander effect, when making allocation decisions. These results are not intended to suggest that LLMs possess human-like cognition; rather, they highlight systematic patterns in model outputs that resemble known human biases.
>
> In human experiments, it was shown the distribution of human responses do NOT exhibit any significant change when participants are making decisions for others as bystanders or in settings where they are one of the players (have skin in the game). While LLMs have been shown to exhibit human-like altruistic behavior in various environments [1][2], we show that they do not always exhibit this behavior in resources allocation settings, since they make self-serving decisions in certain cases.
>
> [1] Qiaozhu Mei, Yutong Xie, Walter Yuan, and Matthew O. Jackson. A turing test of whether ai chatbots are behaviorally similar to humans. Proceedings of the National Academy of Sciences, 121(9):e2313925121, 2024. doi: 10.1073/pnas.2313925121.
>
> [2] Aher, G. V., Arriaga, R. I., & Kalai, A. T. (2023, July). Using large language models to simulate multiple humans and replicate human subject studies. In International conference on machine learning (pp. 337-371). PMLR.
>
> > The paper would benefit from a more in-depth comparison with the growing literature on LLMs and prosocial behavior, economic games, and normative reasoning. While some of this work is mentioned in passing on page 3, the paper misses the opportunity to position its findings in relation to this broader body of research.
>
> We appreciate this feedback and shall add more detail about how our work relates to the broader field of moral alignment in AI and social behaviour of LLMs, in the final version of this paper.

---

> > ### Comment · Reviewer_vXHW · 2025-08-08
> >
> > I still struggle to understand the criteria used for "alignment" beyond a loose concept of how human decisions are not identical to the models. Figure 2 shows very high variance across questions, where for many questions, human subjects do not select EQ with the highest frequency. Even when questions are aggregated (is this the right measure?) humans only select EQ 12.4% of the time. Are the authors saying that anything less than a full replication of ranking and percentages would be a lack of alignment? Or what, for example, would we make of a case in which Table 2 revealed identical percentages in aggregate for HUmans and GPT-4o but when broken down to the specific questions they were not identical for humans and the model? I have no idea how to answer these questions because the authors never laid down the criteria for judging alignment. I think this may have been ok for the first papers in this literature but given that there is quite a large literature of replicating human experiments  on models (and increasing all the time) greater precision is needed in analyzing the results.

---

> > > ### Author Response · Authors · 2025-08-08
> > > **Clarification about "alignment"**
> > >
> > > A key aspect of our work is not merely comparing distributions of responses, but identifying which fairness axioms are prioritized by LLMs versus humans. While we report aggregate behaviors (e.g., Section 3), our main contribution is deeper: **evaluating alignment over normative principles, not just outcomes.**
> > >
> > > This builds on long-standing work in experimental economics examining how humans trade off fairness axioms (e.g., envy-freeness vs. equality) and their interaction with efficiency. Humans often prioritize inequality aversion (EQ) over envy-freeness (particularly in allocations involving money or goods) whereas LLMs rarely do so, revealing a **potential misalignment in underlying normative reasoning.**
> > >
> > > As the reviewer correctly notes, heterogeneity in human responses reflects how perceived fairness shifts across scenarios. Our instances are deliberately designed to capture such trade-offs. While preferences vary, a general tendency toward equitable outcomes is evident. When EQ is not most preferred, it is typically because (i) a purely equitable (EQ*) allocation does not exist (e.g., I3, Fig. 2), or (ii) multiple allocations satisfy both fairness and efficiency (e.g., I5, Fig. 2).
> > >
> > > In all scenarios where EQ* allocations do exist—including those with monetary transfers (Table 4)—LLMs almost entirely neglect them, **systematically overlooking the values of many human decision-makers.** This is shown both at the instance level (Figures 2, 8; Table 4) and in aggregate (Tables 2, 5). The same evidence shows LLMs are more likely than humans to choose allocations maximizing economic efficiency.
> > >
> > > > Or what, for example, would we make of a case in which Table 2 revealed identical percentages in aggregate for HUmans and GPT-4o but when broken down to the specific questions they were not identical for humans and the model?
> > >
> > > Such a model would still be misaligned. If an LLM prioritizes different notions from humans in individual scenarios, it fails to represent human values, even if aggregate numbers match. For example, if humans favor EF in a scenario but the LLM chooses EQ, it is misaligned. This is why we combine **fine-grained instance-level and aggregate analyses**; both support our conclusions.
> > >
> > > We will clarify the above points in the introduction of the paper.

---

### Decision · Program_Chairs · 2025-09-17

**Decision:**

Accept (poster)

**Comment:**

## Summary
The paper seeks to understand the different normative frameworks that appear to govern LLMs responses to questions about allocation of resources. By creating specific allocation scenarios, comparing the responses to what might be produced from different normative frameworks of equity and justice, and evaluating this against human responses, the paper draws a map of "alignment" with human notions of distributive fairness. The authors also weigh on the 'steerability' question by evaluating the effectiveness of different mitigation strategies at changing the default mode of response to one that might lean more towards a particular kind of distributive fairness framework (tldr: it doesn't work well)

## Strengths
* the reviewers found the results interesting, as well as in how the models disagreed.
* the experimental economics framework for evaluation was appreciated.
* the authors put some effort into ensuring prompt robustness.

## Weaknesses
* the primary weakness was around the claims of alignment. It was unclear what the authors were intending to capture via this (mapping to humans? mapping to norms?)
* there were concerns about properly putting this in context of much prior work on prosocial behavior of LLMs.
* there were number of questions around methodology as well.
## Author Discussion
Author discussion was quite active. and some of the reviewers ended satisfied with the author responses and also increased their score slightly.

## Justification
The authors did a good job responding to reviewer comments for the more specific low-level issues. It took them a while to acknowledge the need for a better related work section, and I didn't find that the discussion on the definition of alignment landed in a satisfactory place. In other words, I'm not convinced the paper ended up being any clearer about whether the goal was to see if LLMs mirrored the distribution of human responses, captured a diversity of normative responses, were consistent with each other, or could be steered to any desired outcome. Given the relatively maturity of the field in which this paper is situated, it felt important that the paper address these issues in some form.